# Progeny counter mechanism in malaria parasites is linked to extracellular resources

**Vanessa S. Stürmer[1], Sophie Stopper[1], Patrick Binder[2,3], Anja Klemmer[1], Nicolas P. Lichti[1], Nils B. Becker[3], Julien Guizetti[1] ***

**1** Heidelberg University Medical Faculty, Centre for Infectious Diseases, Heidelberg University Hospital, Heidelberg, Germany, **2** Institute for Theoretical Physics and BioQuant, Heidelberg University, Heidelberg, Germany, **3** Theoretical Systems Biology, German Cancer Research Center (DKFZ), Heidelberg, Germany

\* julien.guizetti@med.uni-heidelberg.de

**Data Availability Statement:** All data is provided in the manuscript or supplemental information.

**Funding:** This work was supported by the Deutsche Forschungsgemeinschaft (DFG)

## Abstract

Malaria is caused by the rapid proliferation of *Plasmodium* parasites in patients and disease severity correlates with the number of infected red blood cells in circulation. Parasite multiplication within red blood cells is called schizogony and occurs through an atypical multinucleated cell division mode. The mechanisms regulating the number of daughter cells produced by a single progenitor are poorly understood. We investigated underlying regulatory principles by quantifying nuclear multiplication dynamics in *Plasmodium falciparum* and *knowlesi* using super-resolution time-lapse microscopy. This confirmed that the number of daughter cells was consistent with a model in which a counter mechanism regulates multiplication yet incompatible with a timer mechanism. *P. falciparum* cell volume at the start of nuclear division correlated with the final number of daughter cells. As schizogony progressed, the nucleocytoplasmic volume ratio, which has been found to be constant in all eukaryotes characterized so far, increased significantly, possibly to accommodate the exponentially multiplying nuclei. Depleting nutrients by dilution of culture medium caused parasites to produce fewer merozoites and reduced proliferation but did not affect cell volume or total nuclear volume at the end of schizogony. Our findings suggest that the counter mechanism implicated in malaria parasite proliferation integrates extracellular resource status to modify progeny number during blood stage infection.

## Author summary

Malaria remains a significant burden on global health and has even seen a resurgence over the last years. The disease is caused by a small unicellular eukaryotic parasite of the Plasmodium genus, which proliferates by invading red blood cells and replicating within them. Contrary to most studied model organisms, Plasmodium does not replicate by binary division, but instead undergoes multiple nuclear division cycles without dividing. Only thereafter those nuclei are packaged into up to 30 new invasive daughter cells. The number of daughter cells per parasite affects the speed of proliferation, but it is currently unclear how it is regulated. This study describes nuclear multiplication in individual parasites in unprecedented detail using super-resolving live cell microscopy. Mathematical

(349355339) to J.G, the Human Frontiers Science Program (CDA00013/2018-C) to J.G, the Daimler und Benz Stiftung to J.G by providing salary for A. K, and the Chica and Heinz Schaller Foundation to J.G. The funders had no role in study design, data collection and analysis, decision to publish, or preparation of the manuscript.

**Competing interests:** The authors have declared that no competing interests exist.

modelling of the data suggests that the parasite uses a 'counter' mechanism to predetermine the final number of daughter cells. By reducing the amount of available nutrients, we show that the parasite modulates its progeny number in response to external cues. These findings suggest that patients with different nutritional status might have varying susceptibility to malaria parasite proliferation in their blood.

## Introduction

Malaria-causing parasites undergo a complex lifecycle with two critical population bottlenecks during transmission from mosquito to human and back. To overcome those bottlenecks the parasite proliferates extensively in the mosquito midgut, the human liver and red blood cells. In these different stages the parasite generates vast numbers of daughter cells within one cell cycle that can range over four orders of magnitude [1]. Malaria pathogenesis, which still kills more than 600,000 people per year [2], occurs during the proliferation of *Plasmodium* spp. in the human blood. After red blood cell (RBC) invasion the parasite generates a variable number of daughter cells, called merozoites, that can range from 12 to 30 [3–6]. Understanding regulatory mechanisms of parasite multiplication is important as the number of merozoites influences growth rate [6]. What limits the number of merozoites is largely unclear.

The division mode of *Plasmodium* spp. is called schizogony and diverges significantly from the classical binary fission observed in most model organisms (Fig 1A). Schizogony entails a particular cell cycle regulation, which proceeds by multiple asynchronous nuclear divisions not interrupted by cytokinesis and leads to a multinucleated schizont stage [7–13]. Only thereafter nuclei are packaged into merozoites during the segmenter stage, which is followed by egress from the host cell [14,15]. The released merozoites invade a new RBC and start the next intraerythrocytic development cycle (IDC). IDC durations vary between *Plasmodium* species but usually constitute multiples of 24 hours. *P. falciparum*, which causes the prevalent and most severe form of malaria, requires 48 h and produces 20 merozoites on average [5]. The IDC of cultured parasites can deviate from this average by a few hours [4,16]. After an initial growth phase, DNA replication, which precedes the first nuclear division, starts at about 30 hours post invasion (hpi) in *P. falciparum* [17,18]. *P. knowlesi*, a zoonotic malaria parasite mostly found in macaques, is a rising threat to malaria eradication and has a shorter IDC of 24 h [19]. *P. knowlesi* was recently adapted for in vitro culture, where it produces about 12 merozoites on average and displays an extended IDC duration of about 27 h [5,20,21].

Few functional studies have quantified effects on merozoite number [6,22–26]. The fact that the parasite can adapt its merozoite number is evidenced by the seminal discovery of a nutrient sensing pathway requiring the KIN kinase, which decreases merozoite number upon calorie restriction in mice [27]. What mechanisms define the number of merozoites and the duration of schizogony remains unclear. In eukaryotes, two phenomenological mechanisms linking cell division to cell growth have been described [28,29]. In a timer mechanism, a cell completes division after a preset time has elapsed. While the interdivision time may be stochastic, it is independent of the growth process. By contrast, in a sizer mechanism a cell completes division once a specific extensive cellular parameter, such as cell size, has reached a set value. Again, the preset size value may be stochastic but is required to be independent of growth speed. In various uni- and multicellular eukaryotes both mechanisms have been described and appear to be implemented by sensing different cellular parameters before engaging in cell division [29–34].

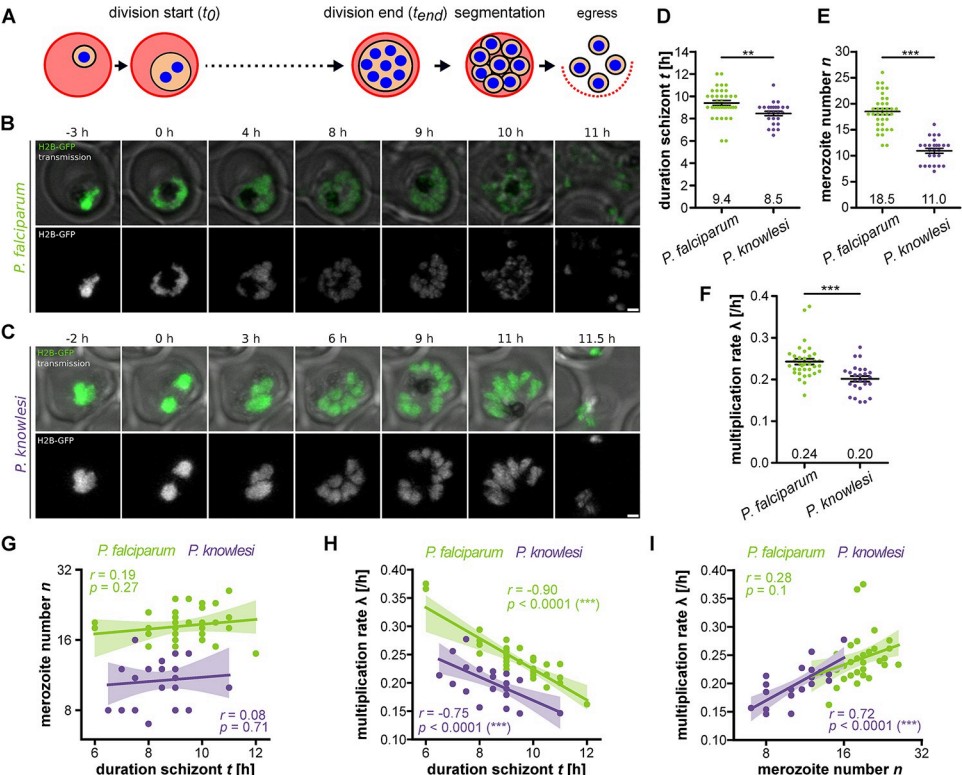

**Fig 1. Duration of schizont stage and resulting merozoite numbers in individual P. falciparum and P. knowlesi parasites. A** Schematic of schizogony during asexual blood stage development **B** Airyscan-processed time-lapse images of *P. falciparum* strain 3D7 episomally expressing H2B-GFP. 0 h indicates first nuclear division. Shown are maximum intensity projections. Scale bars are 1 μm. **C** as in B for *P. knowlesi* strain A1-H.1. **D** Quantification of duration of schizont stage in hours**E** merozoite number and **F** multiplication rates based on *t* and *n*. Error bars show mean and SEM. All statistical analysis: t-test with Welch's correction **G-I** Correlation and regression curves of shown *t*, *λ*, and *n* values for *P. falciparum* (green) and *P. knowlesi* (violet) with data bootstrapped with 95% confidence interval. Given are Pearson correlation coefficient *r* and *p* values. N = 35 for *P. falciparum* and N = 26 for *P. knowlesi* all from three independent replicates.

In the context of multinucleated schizogonic division, the picture is more complex. In addition to cell growth the nuclear multiplication rate and the final number of nuclei, (i.e. merozoites) are relevant system parameters. Analogous to what has been proposed in model eukaryotes we previously put forward two stereotypical and competing models that can contribute to the control of nuclear number i.e. a 'timer' or a 'counter' [5,35].

Differentiating between those models is critical to guide further analysis of parasite proliferation. The timer model posits that a clock-like molecular process tracks the time since the start of nuclear multiplication and stops it when a preset target duration has elapsed. The process is independent of growth, which implies that the nuclear multiplication rate and the target duration vary between parasites in a statistically independent way. A timer would necessitate a biochemical reaction functioning as a molecular hourglass and can arise, e.g., if nuclear multiplication is synchronized and terminated in response to external cues. By contrast, the counter model posits that a molecular process tracks the number of nuclei and stops their multiplication once a target number has been reached. Counting is independent of growth, which implies that the target number and nuclear multiplication rate are statistically independent. A counter would imply a molecular factor limiting proliferation and can arise, e. g., if continued nuclear multiplication successively consumes a scarce resource.

Previous analyses of *P. falciparum* DNA replication and nuclear division dynamics observed that the duration of the first nuclear cycle covaries with overall duration of nuclear multiplication. This excluded a timer and corroborated a counter as the mechanism regulating termination of nuclear multiplication [35]. Data directly reporting a correlation between duration of schizont stage and merozoite number is, however, still missing. It is also unclear which cellular or extracellular parameters are relevant for determining final merozoite number.

Most studies that have quantified schizogony have so far relied on fixed cell or population analysis. Here, we produce time-resolved data of key cellular parameters in individual cells. By using Airyscan-detector-based super resolution live cell imaging modalities we achieve sufficiently low photo-toxicity and high spatial resolution to reliably track live cells throughout schizogony and count individual merozoites. We correlate duration of the schizont stage, cell size, and merozoite number in single cells, which provides evidence for a counter mechanism. We further demonstrate that throughout schizogony *P. falciparum* infringes on the otherwise ubiquitously constant nucleocytoplasmic ratio (N/C-ratio) [36], and find that diluting nutrients from culture media reduces progeny number.

## Results

### Nuclear multiplication dynamics in *P. falciparum* and *knowlesi*

To correlate duration of the schizont stage with merozoite number, we generated *P. falciparum* and *P. knowlesi* strains episomally expressing GFP-tagged Histone 2B (H2B-GFP), which labels nuclear chromatin (Figs 1B, 1C and S1A). Episomal expression did not significantly affect parasite multiplication rate (S2 Fig). To resolve individual daughter nuclei in live cells, we employed Airyscan detector-based microscopy, which improves spatial resolution by a factor of about 1.7 beyond the diffraction-limit, and recorded schizogony from the first nuclear division until egress from the host cell (S1 and S2 Movies). Imaging under hypoxia conditions and in riboflavin-free media improved cell viability and reduced photo-toxicity [37]. Even though the first nuclear division events can be readily detected separating individual nuclei after the first two rounds of division was challenging due to their spatial proximity. The improvements in resolution and 3D image analysis, however, allowed us to count the final number of nuclei routinely and reliably at the transition into the segmenter stage where nuclei are again more spread out. The final number of segmenter nuclei always perfectly matched the number of merozoites observed just before egress. Since it was also shown that every nucleus is packaged into an individual merozoite [14,38], we will from now on refer to this number as merozoite number ($n$). In this study, we define the start of the schizont stage ($t_0$) by the first appearance of two distinct nuclei and its end ($t_{end}$) as the time when the maximal number of nuclei was first reached (Fig 1A). Possibly, other cell cycle transitions point such as invasion could have relevance, although this would require a molecular mechanism monitoring significantly longer time spans. Our selection of $t_0$ is motivated by the understanding that entry into the schizont stage marks an important cell cycle transition point, after which the 'decision' about the final number of nuclei is executed [1,39]. The duration of the schizont stage ($t$), defined as $t = t_{end} - t_0$, for *P. falciparum* was around 9.4 h (Fig 1D). Remarkably *P. knowlesi*, which has a much shorter IDC [18,20], showed an only slightly reduced duration of the schizont stage of around 8.5 h. The merozoite number, however, was much lower, with $n = 18.5$ and $n = 11.0$, in *P. falciparum* and *knowlesi*, respectively (Fig 1E), which matched with the previously measured values in fixed cells [5]. To further validate our assay, we used an alternative marker, the microtubule live cell dye SPY555-Tubulin (S3 Movie, S3A and S3B Fig). Defining the first mitotic spindle extension as $t_0$ and appearance of subpellicular microtubules as $t_{end}$, resulted in very similar and reproducible values for *P. falciparum* and *P. knowlesi* (S3C–S3F

Fig). The dynamics of nuclear multiplication follow an exponential growth function [35]. Merozoite number $n$ is therefore determined by the nuclear multiplication rate ($\lambda$) and duration of the schizont stage ($t$) by $n(t) = 2\ e^{\lambda t}$. The multiplication rate ($\lambda$), which is the exponent determining doubling speed for nuclear number, was lower in *P. knowlesi* than in *P. falciparum* (Fig 1F). We next analyzed correlations between the different proliferation parameters (Fig 1G–1I). When plotting $n$ against $t$ for *P. falciparum* and *P. knowlesi* the slopes of the regression curves did not significantly deviate from zero showing that parasites do not produce more nuclei when spending more time dividing (Fig 1G). The multiplication rate $\lambda$ showed a negative correlation with $t$, which indicates that parasites that multiply nuclei faster reach their final number of merozoites earlier (Fig 1H). Fast-multiplying parasites, however, produced significantly more nuclei on average only in *P. knowlesi* (Fig 1I). These data provide a detailed quantitative description of schizogony dynamics and reveals that *P. knowlesi* and *P. falciparum*, despite their different progeny numbers, display similar correlations.

## Statistical analysis supports counter mechanism

We next aimed to test the proposed models in view of these data. Reducing the data to the observed $t−n$ correlations only, is not sufficient to discriminate between the timer and counter models (Fig 1G–1I), so we undertook a more thorough statistical analysis. The timer model requires $t$ and $\lambda$ to be independent random variables, whereas in the counter $n$ and $\lambda$ are independent (Fig 2A and 2B). As a preliminary observation, we point out that if $\lambda$ were tightly controlled to remain constant across individual cells, both models would predict that the observed $t−n$ pairs lie on the same exponential growth curve (Fig 2A and 2B, black line). This would preclude model discrimination from these data. Fortunately, fluctuations in the growth rate do occur (Fig 1F) and therefore two models make different predictions for the joint distribution of final numbers and durations (Fig 2A and 2B). The timer model constructs marginal distributions for $\lambda$ and for $t$ from data (S4 Fig), and by assuming their independence predicts the joint distribution for $t−n$ pairs (see methods for more details). The counter model assumes independent variables $\lambda$ and $n$ with marginal distributions taken from data, and again predicts the joint distribution for $t−n$ pairs (S4 Fig). We overlayed those model predictions with recorded data from the H2B-GFP expressing *P. falciparum* strain (Fig 2C). By visual inspection, the data are in good agreement with the prediction of the counter (red), but not of the timer (blue). To evaluate model support by data quantitatively we calculated the Bayes factor (odds ratio) $K$ for counter vs. timer, which confirmed that the counter model is strongly and robustly preferred over the timer model ($K = 3.5 \times 10^7$, $K>100$, indicating "decisive" preference, in 96.7% of bootstrap resamples). Moreover, we assessed goodness-of-fit by analyzing Jensen-Shannon distance (JSD) of our data from the predicted counter model density. The JSD was less than the JSDs in a fraction $p = 0.61$ of synthetic data sampled from the counter model. This showed that the counter model indeed fits acceptably (Fig 2D). By contrast, the timer model fits poorly, $p<0.001$. We repeated these analyses for time-lapse data generated with the SPY555-Tubulin labeled *P. falciparum* and found strong support for the counter model ($K = 8.6 \times 10^9$, $K>100$ in 99.8%) (Fig 2E and 2F). For the *P. knowlesi* line expressing H2B-GFP (S3B Fig) the data could not clearly discriminate between the counter and timer (Fig 2G and 2H). While the timer achieved an acceptable fit ($p = 0.176$) and was slightly preferred ($K = 0.04$), this preference was not robust to small-sample variation ($K<1/100$ in 41.3%, but $K>100$ in 7.0%). When analyzing SPY555-Tubulin labeled *P. knowlesi*, the data again showed a clear preference for the counter model ($K = 4.7 \times 10^7$, $K<100$ in 97.4%) (Fig 2I and 2J). Together our data support the counter model for *P. falciparum*. For *P. knowlesi* the results were less discriminating with one out of two dataset showing strong counter support (Fig 2K).

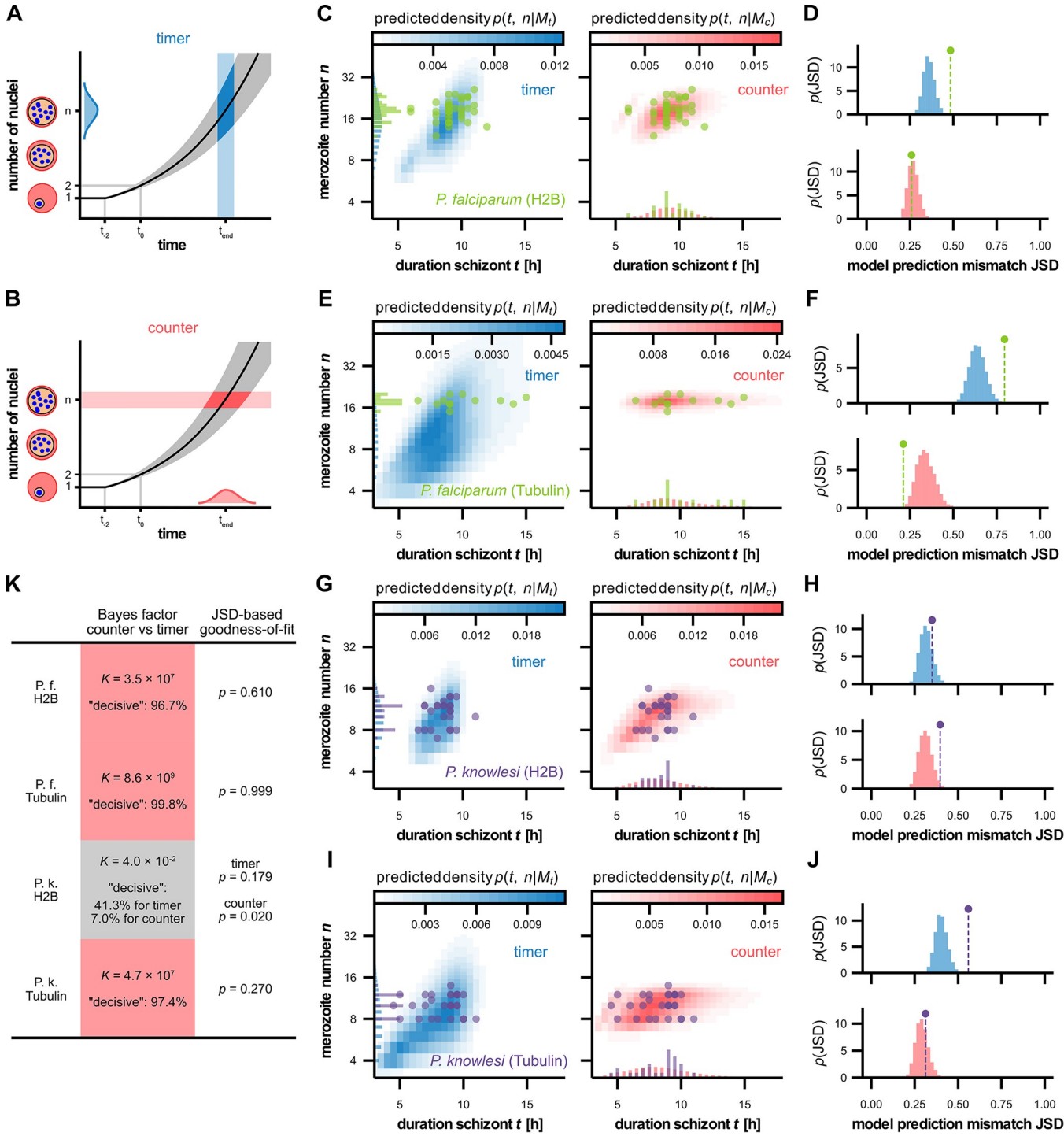

**Fig 2. Predictions by timer and counter models compared to data.** *A In the timer model (blue), exponential nuclear multiplication is stopped after a preset duration t. Growth rate and duration variability entail variable merozoite numbers n. B In the counter model (red), multiplication is stopped after reaching a preset number n. Growth rate and number variability entail variable t. C Measured number vs. duration for P. falciparum H2B-GFP expressing schizonts. Timer (blue, left) and counter (red, right) predict different densities in the t−n plane and marginal distribution in n and t, respectively, based on the data. D Jensen-Shannon distance (JSD) between data and model density (marker) for timer and counter (top and bottom, respectively). JSD histograms for synthetic data sampled from timer (blue) and counter (red) models show that the data are representative of a counter but not timer sample. E-F as (A-B) but for SPY555-Tubulin labelled P. falciparum. G-H as (A-B) but for P. knowlesi H2B. I-J as (A-B) but for SPY555-Tubulin labelled P. knowlesi. K Bayes factors K (red background) show a strong preference for the counter model, which is robust to small sample variation, indicated by high percentages of bootstrap resamples with "decisive" (i.e. K>100) support. P. knowlesi H2B data (gray backgound) show no clear preference. Compare (CEGI). Goodness-of-fit quantified by the probability p to observe synthetic data samples that more distant from the model than the data is (area fraction to the right of the marker in panels (DFHJ).*

The counter model has the distinguishing property that even with strong fluctuations in growth speed, the final nuclear number can be controlled precisely. To demonstrate that this is difficult to achieve within the timer model, we shuffled data by reassigning $\lambda - t$ pairings randomly and then computed resulting merozoite numbers according to a timer (S5 Fig). Indeed, the variability in predicted numbers was increased significantly over that of the data. Assuming a more precise control of merozoite number is beneficial, this provides a rationale for the appearance of a counter controlling *Plasmodium* nuclear multiplication.

### Duration of merozoite formation is constant between *P. falciparum* and *P. knowlesi*

To interrogate the duration of merozoite formation we quantified the time from end of schizont stage ($t_{end}$) until egress from the host cell (Fig 1A). We found it to be identical in both species at around 2.7 h (Fig 3A). The presence of more nuclei did not prolong the time required until egress (Fig 3B), and neither was it correlated with the duration of the preceding schizont stage (Fig 3C). This indicates that the time used for merozoite formation is largely fixed and independent of previous events.

### Merozoite number does not depend on host cell diameter

The counter mechanism raises the question which cellular or extracellular parameters might limit merozoite number. Cell size has been identified as an important parameter linked to the initiation of cell division in several organisms [28,40]. We therefore investigated whether size of the host erythrocyte might influence merozoite number, possibly by providing more nutrients or more space for the parasite. Since RBC volume is challenging to assess directly in our assay, we measured the average RBC diameter in cells lying flat (Fig 4A). Host erythrocytes become much more spherical as a consequence of parasite growth [41]. Due to this rounding effect *P. falciparum*- and *P. knowlesi*- infected RBCs had a smaller diameter compared to uninfected ones at ($t_0$) and ($t_{end}$) (Fig 4B). We found no significant correlation between RBC diameter and merozoite number at defined timepoints (Fig 4C and 4D). Whether this finding extrapolates to a lack of correlation with RBC volume remains to be investigated.

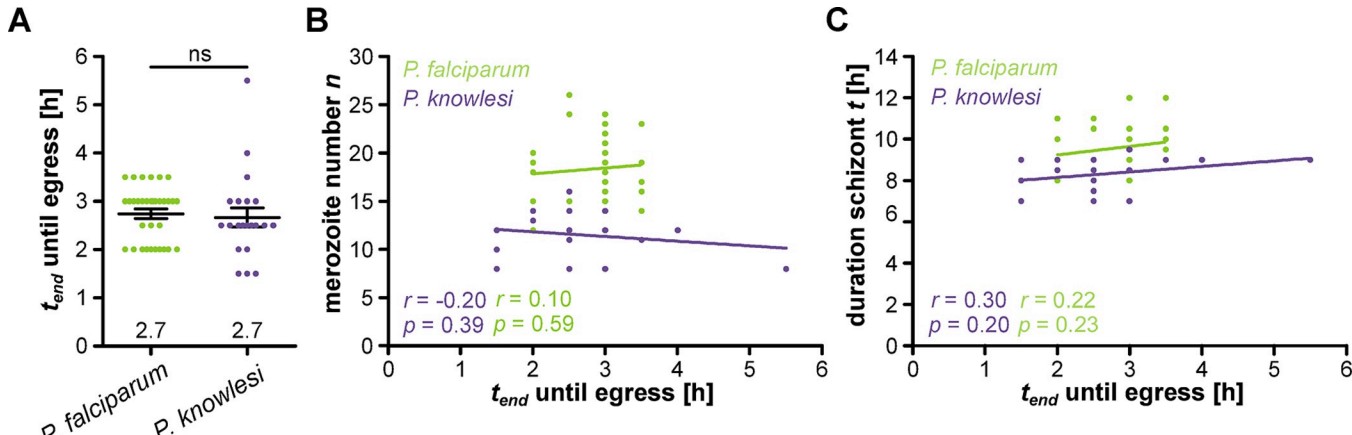

**Fig 3. Time required for merozoite formation does not depend on the number of nuclei.** Quantification of time between schizogony end and egress for H2B-GFP expressing P. falciparum (green) and P. knowlesi (blue) in hours. Given are mean and SEM. Statistical analysis: t-test with Welch's correction. **B** Correlation of merozoite number against segmentation time. **C** Correlation of duration of schizont stage against segmentation time. Given are Pearson correlation coefficient r and p values. N = 33 for P. falciparum and N = 21 for P. knowlesi all from three independent replicates.

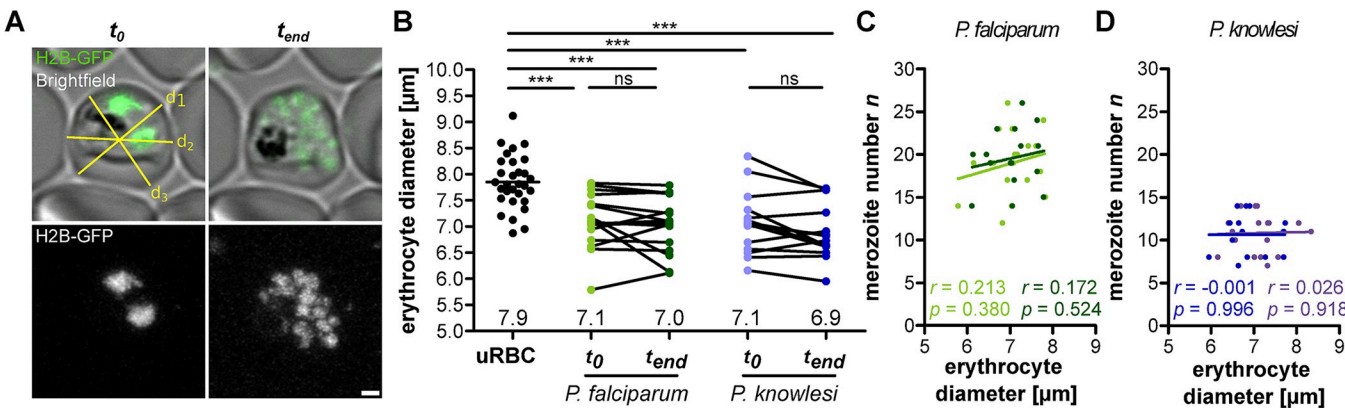

**Fig 4. Erythrocyte diameter has no effect on merozoite number in P. falciparum. A** Airyscan-processed images of P. falciparum 3D7 episomally expressing H2B-GFP. Erythrocyte diameter was measured by averaging three diameters (yellow lines) at different angles. For each biological replicate a different donor blood batch was used. **B** Erythrocyte diameters in uninfected red blood cells (uRBC) and P. falciparum (green) and P. knowlesi (blue) infected red blood cells. Measured as paired values at $t_0$ and $t_{end}$. Means of three diameter measurements for each RBC are plotted. Mean is shown for uRBCs. Statistical analysis: t-test with Welch's correction. **C** Correlation of P. falciparum merozoite number against erythrocyte diameter. **D** As in C for P. knowlesi. Given are Pearson correlation coefficient r and p values. N = 30 for uRBC, N = 19 ($t_0$) and N = 16 ($t_{end}$) for P. falciparum H2B-GFP and N = 18 ($t_0$) and N = 15 ($t_{end}$) for P. knowlesi H2B-GFP all from three independent blood donors.

## Cell size at onset of schizont stage correlates with merozoite number

To measure cellular and nuclear volume of the parasite we generated a *P. falciparum* strain episomally expressing soluble GFP and mCherry tagged with a nuclear localization signal (NLS) (Figs 5A and S1B). We could not produce dual-labeled *P. knowlesi* strains due to the lack of an effective second resistance cassette and therefore continued our analysis focusing on *P. falciparum*. The dual-marker *P. falciparum* strain spent the same time in the schizont stage as the H2B-GFP strain although its merozoite number was slightly higher, which could be explained by the absence of histone tagging that might affect nuclear multiplication (S6A and S6B Fig). Again, the data clearly supported the counter model ($K = 2.6 \times 10^5$, $K > 100$ in 86.7%) (S6C and S6D Fig). We acquired long-term time lapse microscopy data of progression through schizogony (S4 Movie) and applied automated image thresholding on the cytosolic GFP signal (S7 Fig) to quantify parasite cell volume (Fig 5B). These data started up to 7 h before schizont onset and showed a steady increase in total cell volume from ~20 $\mu m^3$ to ~80 $\mu m^3$ around the end of schizogony (Fig 5C). This aligned with previously reported and modelled values of 18–32 $\mu m^3$ at 30 to 34 hpi and 60–76 $\mu m^3$ at 48 hpi [41–43]. Importantly, we wanted to test whether parasite cell volume did correlate with $n$, $\lambda$, and $t$. For analysis we identified three relevant timepoints $t_{-2}$, $t_0$, and $t_{end}$. The $t_{-2}$ time point corresponds to the onset of DNA replication which occurs on average 2 h before the first nuclear division [35]. We found a significant correlation of merozoite number with the cell volume around the onset of schizogony (Fig 5D and 5E), while merozoite number did not correlate with the cell volume at the end of schizogony (Fig 5F). Consistently, we found a correlation of $\lambda$ with cell size around the onset of schizogony, while duration of schizont stage was not correlated (S8 Fig). Taken together this suggests that the setting of the counter is linked to parasite cell size.

## Nuclear compaction during schizogony does not offset increase of N/C-ratio

Quantification of total nuclear volume was performed by manual thresholding at $t_{-2}$, $t_0$, and $t_{end}$, as the decrease of signal over time prevented automated thresholding (S7 Fig). We

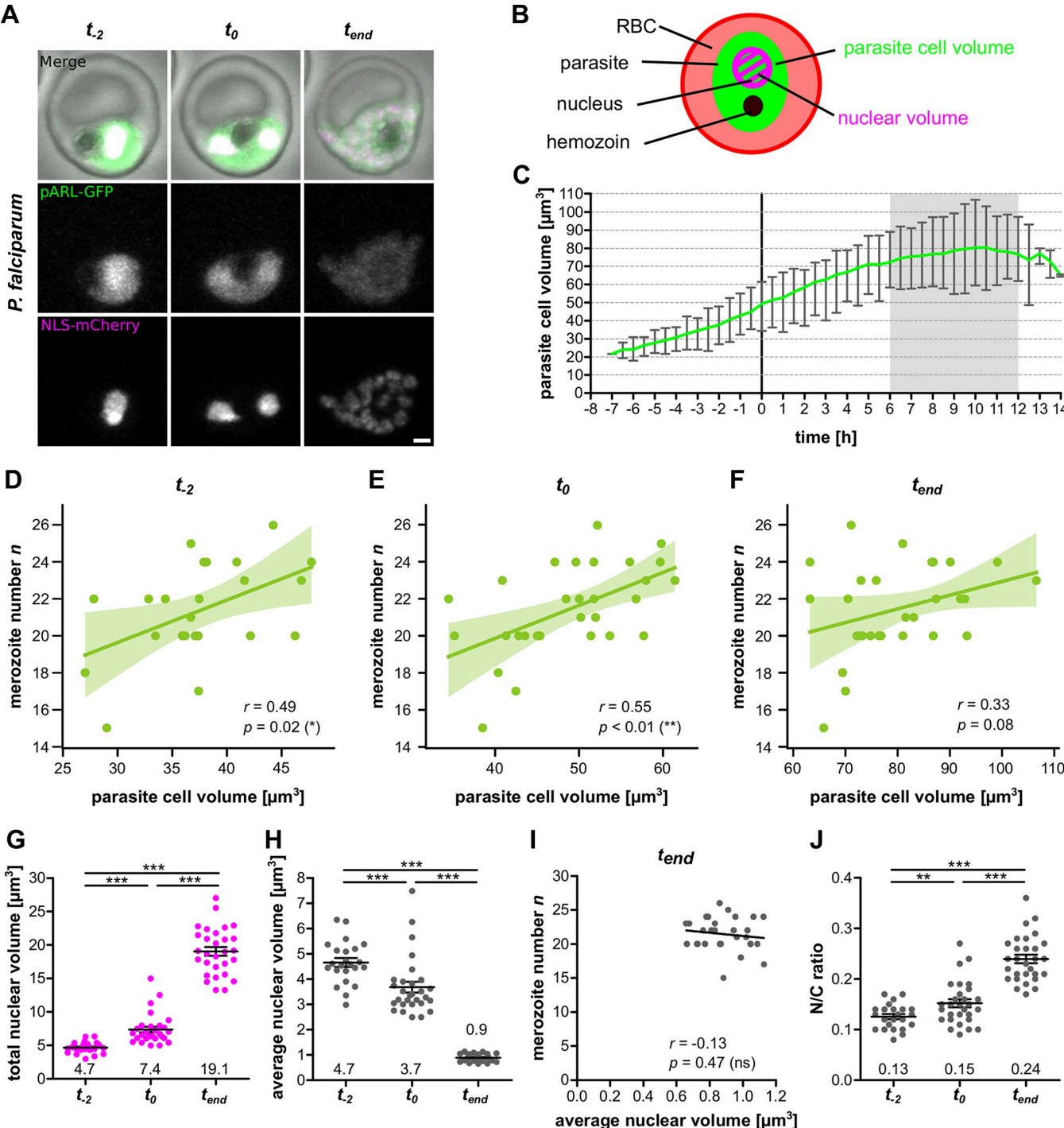

**Fig 5. Cell size around schizont stage transition correlates with merozoite number and N/C-ratio increases.** *A Airyscan-processed time-lapse images of P. falciparum 3D7 episomally expressing cytosolic GFP (pARL-GFP, green), nuclear mCherry (NLS-mCherry, magenta). Selected timepoints were pre-schizogony ($t_{-2}$), schizogony start ($t_0$) and schizogony end ($t_{end}$). Maximum intensity projections are shown. Scale bar is 1 μm. B Schematic for segmentation of parasite cell volume (green) and nuclear volume (magenta). C Quantification of parasite cell volume over time. Line represents mean and error bars range. Curves were aligned to schizogony start (0 h). Grey zone indicates range of measured schizont stage durations. D Correlation of merozoite number against cell volume at pre-schizogony ($t_{-2}$), E schizogony start ($t_0$) and F schizogony end ($t_{end}$). Given are Pearson correlation coefficient r and p values. Values are bootstrapped to 95% confidence interval G Total nuclear volume for individual cells at $t_{-2}$, $t_0$, and $t_{end}$. H Average nuclear volume (total nuclear volume divided by nuclear number) for individual cells. I Merozoite number against average nuclear volume at end of schizogony ($t_{end}$). Given are Pearson correlation coefficient r and p values. J Total nuclear volume divided by cell volume (N/C ratio) for individual cells. All error bars represent mean and SEM. Statistical analyses: t-test with Welch's correction. N = 23 for $t_{-2}$ and N = 29 for all others from three independent replicates.*

observed an about 4-fold increase in total nuclear volume throughout the schizont stage (Fig 5G). Simultaneously the average volume of individual nuclei decreased from 4.7 μm$^3$ to 0.9 μm$^3$ at the end of schizogony. This indicates that the parasite compacts its nuclei to accommodate their rising numbers within a limited cellular volume. The nuclear volume at $t_{end}$ had a notably low variability (Fig 5H), and the reduction in nuclear volume did not scale with the total number of nuclei (Fig 5I), which suggests that the final nuclear volume might represent a state of maximal compaction. Despite nuclear compaction the overall N/C-ratio, which has been described to be universally constant across eukaryotic cell cycles [36], almost doubled from 0.13 to 0.24 (Fig 5J), but did not correlate with final nuclear number (S9 Fig). We also tried to approximate nuclear volume in *P. knowlesi* by manual segmentation of the H2B-GFP signal (Fig 1C). This showed trends similar to *P. falciparum* although the minimal nuclear volume was higher and the merozoite number showed a slight anticorrelation with average nuclear volume due to two outlier cells with particular large nuclei (S10 Fig). Taken together this suggests that the nuclear compaction we document is not a gradual response to the density of nuclei but rather a convergence to a uniformly small nuclear size.

## Nutrient-limited conditions reduce merozoite number

Parasites take up resources for growth from the host red blood cell and the surrounding medium. To test whether nutrient status of the medium affects proliferation we experimented with several dilutions of complete RPMI cell culture medium with physiological 0.9% NaCl solution (S11 Fig). This reduces every media component while maintaining osmolarity and pH. While dilutions to 0.33x or less ultimately caused parasite death, they could be grown over prolonged periods in 0.5x diluted medium (S11 Fig). Hematology analysis showed that of all RBC indices only mean corpuscular volume (MCV) differed by about 1.2% indicating that host cell health is maintained (S12 Fig). Aside a lower parasitemia (Fig 6A), we could observe a significant reduction in parasite multiplication rate compared to normal 1x culture media condition (Fig 6B). This was accompanied by a reduced merozoite number as measured by 3D imaging of fixed and Hoechst-stained wild type segmenters (Fig 6C). Using time-lapse microscopy, we imaged the schizont stage of the cytoplasmic GFP and nuclear mCherry-expressing parasite strain that were switched to 0.5x medium during the preceding ring stage (S5 Movie). Duration of schizont stage was slightly longer and the significant reduction in merozoite number was reproducible (Fig 6D and 6E). This resulted in a lower multiplication rate (Fig 6F), while the data remained fully compatible with the counter model ($K = 3.6 \times 10^8$, $K > 100$ in 99.5%) (Figs 6G, 6H and S13).

## Nutrient-limited conditions increases average nuclear volume

We also quantified cell volume in 0.5x media conditions and only found a small increase at $t_{-2}$ and $t_0$ while at $t_{end}$ cell volumes were similar (Fig 7A and 7B). We, however, did not observe a correlation between cell volume and merozoite number at early schizont stages anymore (Fig 7C and 7D), while at $t_{end}$ a slight correlation appeared (Fig 7E). This shows that even though cell size can correlate with merozoite number in standard growth conditions (Fig 5D and 5E) the division counter is not strictly coupled to it. A notable difference was the total nuclear volume being higher in 0.5x medium cells at the onset of nuclear division (Fig 7F). Remarkably, at the end of division it was identical to standard conditions despite the lower final number of nuclei. This translated to a significantly higher average nuclear volume (Fig 7G), which was, contrary to standard media conditions, anticorrelated with merozoite number (Fig 7H). Finally, this lower number of nuclei with a higher volume in an equally big cell yielded the same N/C-ratio at $t_{end}$ (Fig 7I). Despite a resource-limited environment and reduced

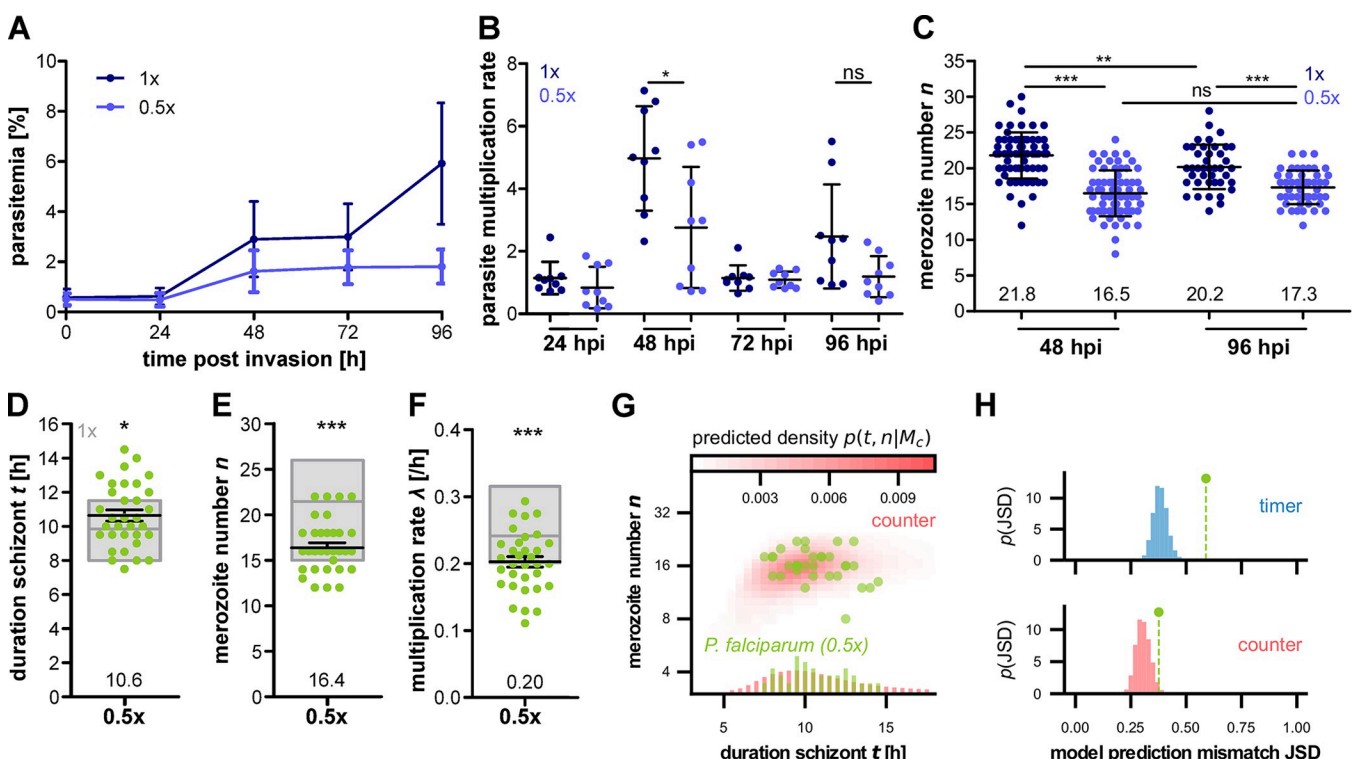

**Fig 6. Medium dilution causes reduction in merozoite number in P. falciparum. A** Growth curve of synchronized 3D7 WT grown in normal 1x (dark blue) and diluted 0.5x (light blue) medium starting as early rings. Starting parasitemia was set around 0.2%. Medium was changed every 24 h. Plotted are mean and SD. **B** Multiplication rate from A. **C** Merozoite numbers counted by fluorescent microscopy after 48 and 96 h. **D** Duration of schizont stage t in hours, **E** merozoite number n and **F** multiplication rate λ [/h] of P. falciparum 3D7 episomally expressing cytosolic GFP and nuclear mCherry in 0.5x diluted medium. Parasites were placed in medium conditions 24–30 hours prior to imaging. Gray boxes show mean, min and max of previously presented data in 1x conditions (S3 Fig). Statistical analyses: t-test with Welch's correction. N = 3 from three independent replicates for A-B, N = 67 (1x, 48h), 69 (0.5x, 48h), 41 (1x, 96h) and 50 (0.5x, 96h), and N = 33 from ten independent replicates for D–H. **G** Measured number vs. duration for P. falciparum H2B-GFP expressing schizonts. Counter (red) predict specific densities in the t−n plane and marginal distribution in n and t, respectively, based on the data. **H** JSD between data and model density (marker) for timer and counter (top and bottom, respectively).

merozoite number the parasites still converge towards an equivalent total nuclear volume, cell volume, and N/C-ratio (Fig 7B, 7F and 7I). Since these cellular parameters are interlinked, it remains to be seen, which one might be dominant.

## Discussion

In this study we quantify key biophysical cell division parameters to correlate duration, cell volume, and nuclear volume with the number of daughter cells generated. Our findings support previous models which propose that parasites use a limiting factor to regulate proliferation [5,35]. We also suggest extracellular resources might be linked to modulation of parasite progeny number. While multiplying the nuclei get compacted which leads to an atypical increase in N/C-ratio (Fig 8). An unexpected finding was that the schizont stage in *P. knowlesi* had a duration similar to *P. falciparum* despite its IDC being much shorter. Recently, the schizont stage duration was quantified at about 30% of IDC duration in *P. knowlesi* and *P. falciparum* [18]. For *P. knowlesi* the authors found the duration from the first DNA replication, detected by appearance of nuclei pulse-chase labeled with Ethyl-deoxyuridine (EdU), to the peak of median nuclear number in fixed cell populations to be around ~9–11 h. We found 8.5 h on average for the duration of first to last nuclear division, which matches well when adding

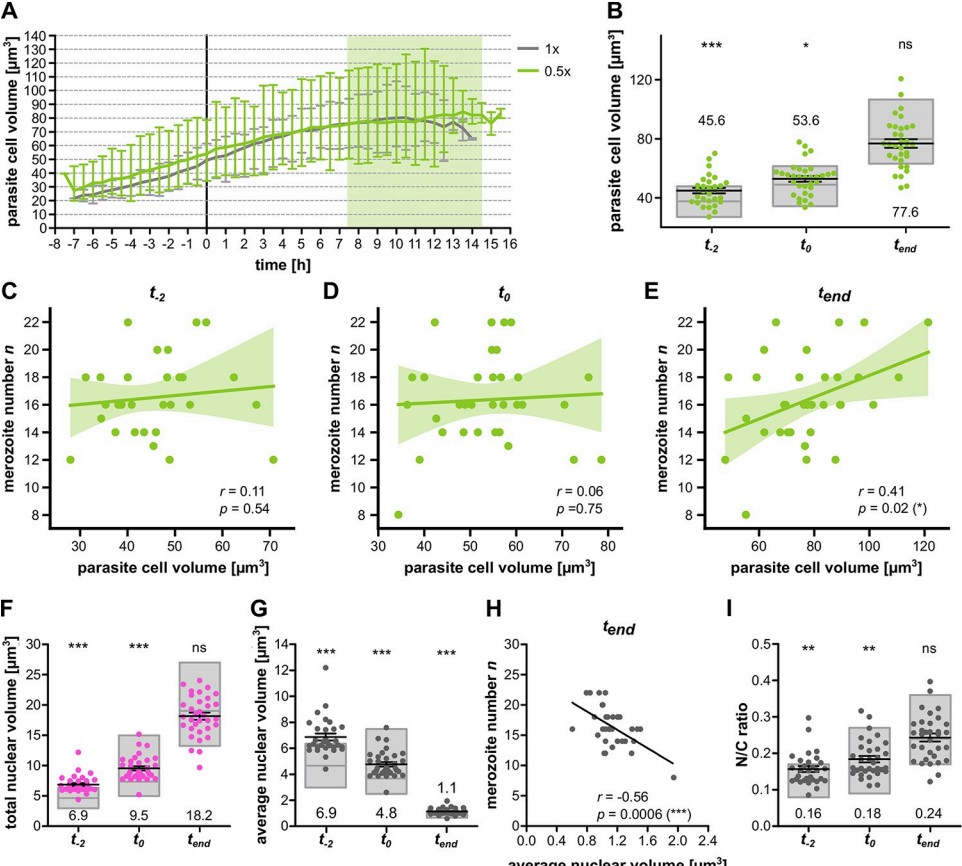

**Fig 7. No correlation between cell volume around schizogony start and merozoite number but preserved N/C ratio increase under diluted medium condition. A** Quantification of parasite cell volume over time of time-lapse imaging with P. falciparum 3D7 episomally expressing cytosolic GFP and nuclear mCherry in 0.5x diluted medium with 0.9% NaCl. Parasites were placed in diluted medium conditions 24–30 hours prior to imaging and late trophozoites were selected for movies. Line represents mean and error bar range; grey line shows data acquired in 1x medium (Fig 5). Curves were aligned to schizogony start (0 h). Light green zone indicates range of measured schizont stage durations under diluted conditions. **B** Comparison of parasite cell volume from A between 0.5x medium (green dots) and 1x medium (grey boxes showing mean, min, max) at pre-schizogony ($t_{-2}$), schizogony start ($t_0$) and schizogony end ($t_{end}$). **C** Correlation of merozoite number against cell volume at pre-schizogony ($t_{-2}$), **D** schizogony start ($t_0$) and **E** schizogony end ($t_{end}$). Given are Pearson correlation coefficient r and p values. Three cells shown for $t_{end}$ and $t_0$ were not imaged at $t_{-2}$ **F** Total nuclear volume for individual cells at $t_{-2}$, $t_0$, and $t_{end}$. **G** Average nuclear volume (total nuclear volume divided by nuclear number) for individual cells at timepoints like in F. **H** Merozoite number against average nuclear volume at end of schizogony ($t_{end}$). Given are Pearson correlation coefficient r and p values. **I** Total nuclear volume divided by cell volume (N/C ratio) for individual cells at timepoints described in F. All error bars represent mean and SEM. Statistical analyses: t-test with Welch's correction. N = 30 for $t_{-2}$ and N = 33 for all others from ten independent replicates.

the 2 h required for the first round of DNA replication [35]. For *P. falciparum*, however, the authors found ~15 h, while we estimate the equivalent duration at 11.4 h. Strain to strain variation as well as the different methods used, i.e. time-lapse imaging vs pulse-chase labeling of Thymidine Kinase overexpressing parasites, could explain these differences. Furthermore, the authors indicated a time lag of about 10 h (32–42 hpi) between the time point when the first and the last *P. falciparum* parasite starts replication, which limits the temporal resolution of their assay. For their *P. falciparum* strain the authors further indicate an IDC duration of 48 h, while we estimate our strain to be slightly faster. *P. knowlesi* on the other hand displays a slightly increased IDC duration of about 27 h after in vitro adaptation [20]. In general, while

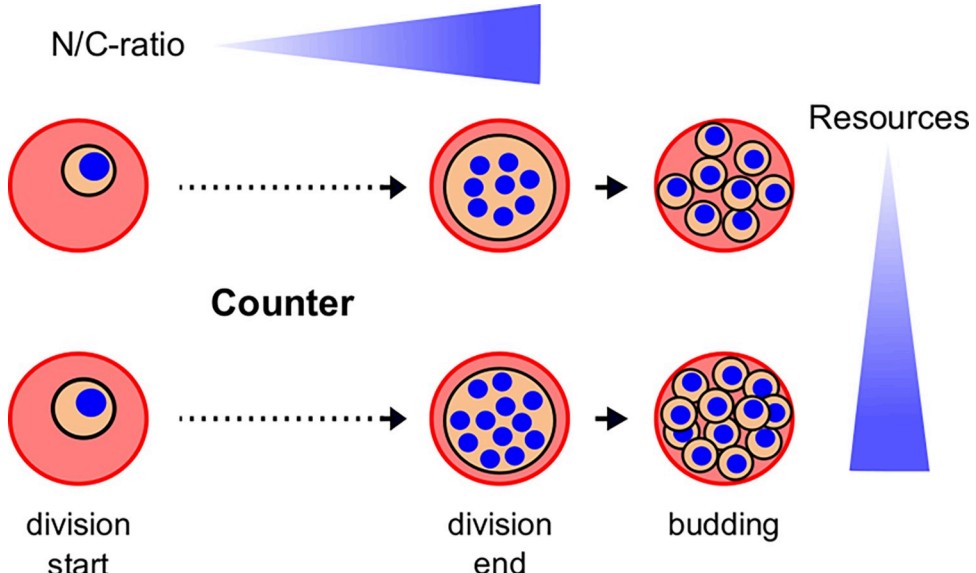

**Fig 8. Schematics of nuclear division events leading to high or low numbers of merozoites.** As Plasmodium parasites undergo nuclear division, they increase their N/C-ratio. High merozoite numbers can result from larger cells at onset of division but are more clearly linked to nutrient concentration in the cell culture medium.

relative proportions and correlations of the cell cycle phases are informative, absolute durations have to be considered with some caution in vitro.

Why *P. knowlesi* produces fewer daughter cells in a similar time remains unclear, but one can note that its merozoites are significantly bigger [44], which might limit their number. Among other Plasmodium species, *P. berghei* produces ~12, while *P. chabaudi* only generates ~6–8 merozoites within their 24 h long IDC [45]. *P. malariae*, whose IDC lasts 72 h, generates only around ~8–10 merozoites per infected RBC, so overall no correlation appears to exist between progeny number and IDC duration across species. For *P. falciparum* and *P. knowlesi* we, however, consistently observe that the number of merozoites is independent of the time spent on nuclear divisions. Despite the simple dichotomous nature of the timer and counter models we find good agreement of the counter with the data for all lines and conditions tested for *P. falciparum*, as has been shown by a previous study analyzing DNA replication dynamics in *P. falciparum* [35]. The observation that in *P. knowlesi* one dataset favors the counter while the other is less decisive indicates that growth regulation is certainly more complex and might occur in different regimes than what our models can reflect. As we quantify the dynamics of more cellular parameters we can expand models for the regulation of progeny number. The fact that other multinucleated organisms like the Ichthyosporean *Sphaeroforma arctica* undergo regularly timed nuclear division events that are independent of cell size and growth emphasizes that a counter is not self-evident [33]. Implementing a counter mechanism requires a limiting factor that sets the growth boundary for the system, which we can now attempt to uncover in more detail.

The concept that nutrients are limiting progeny number has already been shown by studying parasite infection in calorie restricted mice [27]. Limiting proliferation by nutrients might provide advantages for the parasite in being able to quickly notice and adapt to changing environmental conditions [46]. KIN was identified as regulator mediating between nutrient sensing and transcriptional response. Which specific nutrients impact counting is of particular interest and a recent study highlights isoleucine and methionine as important factors

influencing merozoite numbers as part of a starvation response pathway [47]. Even though KIN is a divergent kinase it seems to have analogous functions to the TOR or energy sensing pathway, which links nutritional status to cell growth [48]. Such a link is partly coherent with our finding that cell size around the start of schizogony predicts progeny number, although other cellular factors than cell size must be implied in the setting of the counter. How much the artificial cell culture growth conditions used in this reflect parasite regulation in a physiological context is unclear. In the green alga *Chlamydomonas reinhardtii*, mother cell size is linked to the number of uniform-sized daughter cells [31] and is even influenced by light exposure [49]. Number of nuclear division events and daughter cell size are dependent on the amount of CDKG1 (D-cyclin-dependent RB kinase) in the mother cell [31]. How cells can sense cell size has been shown in Xenopus where progressive sequestration of importin alpha to the plasma membrane during cell division is used for measuring cell surface-area-to-volume ratios, thereby regulating nuclear scaling [50].

As eukaryotic cells grow their nuclear volume increases to keep the N/C-ratio constant over time, although it is not clear why. The "Kern-Plasma-relation theory" had already been proposed by Hertwig and Boveri at the beginning of the 20$^{th}$ century [51] and has been corroborated in uni- and multicellular eukaryotes ranging from mammals to plants and fungi [36], which includes multinucleated organisms like *Ashbya gossypii* [52]. Hence, our observation that malaria parasites do not maintain their N/C-ratio constant was surprising. A key difference, however, to most other studied organisms is that parasite growth is limited by the host cell of whose volume it usually occupies up to 80% at the end of development [41]. In this context it might be a better strategy to infringe on the N/C-ratio to maximize the number of daughter cells that can be fitted in the host cell. Another strategy could be the further reduction of nuclear volume, which is usually not determined by DNA content [36]. Our observation that Plasmodium nuclei converge to a small but very uniform volume of about 0.9 μm$^3$ might however suggest a lower limit for nuclear size imposed by the degree to which DNA can be compacted. Coherently the relative cell size of the parasite does not correlate with the relative DNA content as it does in most other cells, which generates specific challenges with respect to gene dosage [53]. On the other hand, we saw in *P. knowlesi*, which has a similarly sized genome, that the minimal nuclear size was 1.5-fold bigger. An alternative explanation is also suggested by our nutrient depletion experiments where despite producing fewer nuclei the N/C-ratio is increasing to the same level. In merozoites themselves, whose volume was measured at about 1.7 μm$^3$ [54], the nuclei occupy an even more significant proportion of the entire cell [15,55]. Hence, the increase of the N/C-ratio during schizogony could also be viewed as gradual progression towards generating a highly compacted cell stage specialized for invasion.

The discoveries presented here were made possible by imaging new division markers with super-resolution time-lapse microscopy, while mitigating the effects of light-induced stress. Overall, this study highlights that resolving single-cell dynamics of cellular processes over longer time scales must gain importance in the study of malaria parasite proliferation [35,56–58].

The quantitative and dynamic analyses carried out here provide important insights into the regulation of progeny number and provides a biophysical and cell biological framework for further quantitative analysis of the rapid proliferation of malaria-causing parasites in the blood.

## Materials and methods

### Parasite cultivation and growth curves

All *P. falciparum* cell lines were cultured in 0+ erythrocytes in cRPMI (RPMI 1640 containing 0.2 mM hypoxanthine, 25 mM HEPES, 0.5% Albumax and 12.5 μg/mL Gentamycin) and

maintained at a hematocrit of 2.5% and a parasitemia of 1–5%. Dishes were kept incubated at 37˚C with 90% humidity, 5% $O_2$ and 3% $CO_2$. *P. knowlesi* cell lines (WT kindly provided by Robert Moon [20,21]) were cultured as described above but cRPMI was additionally supplemented with 10% horse serum. For diluted media conditions the cRPMI as described above was diluted with 0.9% NaCl solution to the indicated final medium concentration and afterwards sterile filtered. Parasite growth for all growth curves was determined using Giemsa-stained thin blood smears. At least 1000 Erythrocytes or 20 parasites were counted for calculating the parasitemia. Double infections were counted as single infections. Parasites were provided with fresh media every 24 or 48 h for unsynchronized and synchronized growth curves respectively. Synchronization to a window of 3–4 h was achieved using pre-synchronization with 0.5% sorbitol followed by MACS purification and reinvasion.

## Cloning of pARL-H2B-GFP and pPkconcrt-H2B-GFP

To generate pARL-H2B-GFP, pARL-GFP was digested with XhoI and AvrII for linearization. PfH2B was amplified from gDNA with forward primer `TTCTTACATATAActcgagATGGTATCAAAAAAACCAGC` and reverse primer `GAACCTCCACCTCCcctaggTTTGGAGGTAAATTTTGTTA`. Ligation was performed using Gibson Assembly (NEB, E2611S) (Gibson et al., 2009) and the resulting plasmid was checked using Sanger sequencing.

To generate Pkconcrt-GFP, an episomal construct for expression of GFP-tagged proteins, we used Pkcon-GFP (kindly provided by Robert Moon) which was digested with XmaI and NotI-HF to cut out the PkHSP70 promoter. PkCRT promoter was amplified from gDNA with forward primer `TATAGAATACTCGCGGCCGCGGTAACGGTTCTTTTTCTGA` and reverse primer `CCCTTGCTCACCATCCCGGGTGTGTGTTTGTTTTAGCGGTG`. Ligation was performed using Gibson Assembly and the resulting plasmid was checked using Sanger sequencing with forward primer `CACAGGAAACAGCTATGACC` and reverse primer `GGTGGTGCAGATGAACTT`. To generate the pPkconcrt-H2B-GFP plasmids for episomal expression of PkH2B, pPkconcrt-GFP was linearized using XmaI and PkH2B was amplified from gDNA with forward primer `TAAAACAAACACACACCCGGGATGGTATCCAAAAAGCCAGCG` and reverse primer `CATAGAACCTCCACCTCCCCTAGGCTTGGAGGTAAATTTCGTAACGGC`. Ligation was performed using Gibson Assembly. Resulting plasmids were checked by a control digest using XmaI and AvrII and sequenced using Sanger sequencing with forward primer `CGATGCGGTAGAAGAGTCGT` and reverse primer `CTTGTGGCCGTTTACGTCGC`.

## *P. knowlesi* transfection

Parasites were synchronized prior to transfection by using 55% Nycodenz cushions and reinvasion assay. 20 μg of plasmid per reaction was precipitated using 1/10 vol. 3M sodium acetate and 2 vol. cold 100% ethanol, washed with cold 70% ethanol and resuspended in sterile TE buffer. For transfection segmenters were enriched using 50 mM cyclic GMP-dependent protein kinase (PKG) inhibitor ML10 and subsequently transfected by electroporation. Selection for plasmid was done using 2.5 nM WR99210 (Jacobus Pharmaceuticals) 24 hours after transfection.

## *P. falciparum* transfection

pARL-H2B-GFP was transfected into 3D7 WT parasites by electroporation of sorbitol synchronized ring-stage parasites and pARL-GFP (kindly provided by Jude Przyborski) into parasites already episomally expressing NLS-mCherry (previously provided by Markus Ganter [59]) with 50–100 μg of purified plasmid DNA (QIAGEN). Plasmid precipitation was done

prior to transfection as stated above. To select for the plasmid, we used 2.5 nM WR99210 (Jacobus Pharmaceuticals) 24 hours after transfection.

## Preparing cells for live cell imaging

For live cell imaging, resuspended parasite culture was washed twice with pre-warmed incomplete RPMI (iRPMI) i.e. without Albumax, and seeded on μ-Dish (ibiTreat, 81156) or μ-Dish 35 mm Quad (ibiTreat, 80416) for 10 min at 37˚C similar as described before [57,60]. Cells were washed with iRPMI until a monolayer remained. For H2B-GFP movies μ-Dishes were filled completely with imaging medium, meaning phenol red-free RPMI1640 with stable Glutamate and 2 g/L $NaHCO_3$ (PAN Biotech, P04-16520) otherwise supplemented like stated above and previously equilibrated in the incubator beforehand for several hours. Dishes were sealed with parafilm and kept at 37˚C prior of imaging. For all other movies 600 μL per well of imaging medium lacking Riboflavin (Biomol, R9001-04.10) but otherwise supplemented to the same composition as stated above was added in the 35 mm Quad dish and the lid was placed loose on top, and the imaging chamber was incubated with 5% $0_2$ and 5% $CO_2$ during the complete imaging period. In case of *P. knowlesi* movies 10% horse serum was added additionally to the imaging medium. For live-Tubulin staining SPY555-Tubulin or SPY650-Tubulin (Spirochrome) was added to the imaging medium to a final concentration of 500 nM. For diluted media conditions cells were transferred into 0.5x media dilution 24–30h before imaging, and live cell imaging was performed in 0.5x imaging medium (imaging medium diluted as described above) selecting for late trophozoites to ensure altered medium conditions since early ring stage for respective parasites.

## Super-resolution live cell imaging

Live cell imaging was performed using point laser scanning confocal microscopy on a Zeiss LSM900 microscope equipped with the Airyscan detector using Plan-Apochromat 63x/1,4 oil immersion objective. The imaging chamber was incubated at 37˚C and 5% $0_2$ and 5% $CO_2$ in a humidified environment. Images were acquired at multiple positions using an automated stage and the Definite Focus module for focus stabilization. Images were taken at an interval of 30 min– 1 hour for a total period of around 20 hours. Multichannel images were acquired sequentially in the line scanning mode using 488 nm, 561 nm, and 640 nm diode lasers at 0.1% laser power except for volume movies at 0.3%. Brightfield images were obtained from a transmitted light PMT detector. GaAsP PMT and Airyscan detectors were used with the gain adjusted at 500–900V. Image size was typically 20.3 x 20.3 μm with a pixel size of 0.04 μm. Z-stack slices were imaged at an interval of 0.35 nm for a total range of 6–7 μm. Subsequently, ZEN Blue 3.1 software was used for the post 3D Airyscan processing with automatically determined default Airyscan Filtering (AF) strength.

## Image analysis and statistics

Image analysis was performed with Fiji. Counting of nuclei was done manually in 3D (z-stack) using the segmenter stage where nuclei were separated the most. Start of schizogony was determined when two separate nuclei were visible for the first time. End of schizogony was determined by no more appearing nuclei and beginning of segmentation. Merozoite number was confirmed by blinding. Movies were acquired in three or more independent replicates for all experiments. Prism5 was used for all statistical analysis. The r-values indicate how well x and y correlate. The p-values of correlation analysis indicate whether the slope of regression curve is significantly non-zero. T- tests were performed for statistical tests between two conditions and in case of unequal variances with Welch's correction. To determine the range of regression

slopes compatible with data in Fig 1G, 1H and 1I, we performed bootstrap re-sampling from the data, and report the resulting (2.5, 97.5) percentile range of linear regression slopes. Data shuffling. To investigate the importance of λ-t correlations appearing we randomly reassigned the (λ,t) data points, generating samples of uncorrelated synthetic data; for each such data point we calculated a merozoite number as predicted by a timer mechanism, shown in S5 Fig.

### Volume determinations

Volume determinations were performed using Fiji. For cellular volume, GFP signal in all z-slices per timepoints was thresholded using the automated thresholding method "RenyiEntropy". Particles with a smaller area than 0.05 $\mu m^2$ were excluded as well as holes to eliminate the area occupied by hemozoin. The volume was calculated by the slicing method approximating a solid volume by slicing it in regular intervals, adding up areas of all sections and multiplying it with the slicing interval. Therefore, the total GFP area per timepoint was multiplied with the z interval of 0.35 μm for calculating the final volume. Image analysis macro used for processing is provided (S1 Text). For nuclear volume, mCherry signal was manually thresholded at timepoints $t_{-2}$, $t_0$ and $t_{end}$ by setting a visual minimal intensity value. The subsequent volume analysis was done as stated above.

### Automated hematology analysis of erythrocytes

0+ erythrocytes have been kept in different medium dilutions (1x, 0.5x, 0.33x and 0.25x) at a hematocrit of 5% for 24 and 48 h prior to automated determination of red blood cell indices using a Sysmex XP-300. Hematocrit was adjusted to physiological levels before analysis. All conditions have been tested in triplicates with blood from one donor.

### Merozoite counting in fixed segmenters

Tightly synchronized late-stage parasites grown in 1X and 0.5x medium since early rings were collected at the end of the first and second cycle and enriched in segmenter stages using the cyclic GMP-dependent protein kinase (PKG) inhibitor ML10 at a final concentration of 25 nM for 2–3 hours. Cells were seeded on imaging dishes and fixed with 4% paraformaldehyde for 20 min at 37˚C. Nuclei were stained using Hoechst. Imaging of individual segmenters was performed using point laser scanning confocal microscopy on a Zeiss LSM900 microscope equipped with the Airyscan detector using Plan-Apochromat 63x/1,4 oil immersion objective. Images were acquired in the line scanning mode using 405 nm diode laser at 0.3% laser power. Brightfield images were obtained from a transmitted light PMT detector. GaAsP PMT and Airyscan detectors were used with the gain adjusted at 500 – 900V. Image size was 10.1 x 10.1 μm with a pixel size of 0.04 μm. Z-stack slices were imaged at an interval of 0.14 nm for a total range of 8 μm. Subsequently, ZEN Blue 3.1 software was used for the post 3D Airyscan processing with automatically determined default Airyscan Filtering (AF) strength. Merozoites were counted by eye using Fiji and confirmed by blinding. Multiple infections were excluded. The experiment was performed in three technical replicates, counting a minimum of 20 segmenters per replicate.

### Mathematical modeling of nuclear multiplication

Nuclear multiplication is exponential to a good approximation [35], so we considered the exponential growth rate λ as a basic variable in addition to the merozoite number *n* and duration *t*. We compared two minimal models for growth termination. In the timer model growth occurs at rate λ until terminated after *t*, where λ and *t* are taken to vary independently between

schizonts. In the counter model growth occurs at rate $\lambda$ until terminated at final number $n$, where $\lambda$ and $n$ vary independently. To construct either model from a given dataset $\{(t_i, n_i)\}$, we first determined a kernel density estimates (KDEs) for the marginal distributions $p(t)$ and $p(n)$ obtained from data and $p(\lambda)$ obtained from observed growth rates $\{\lambda_i = \log(n_i/2)/t_i\}$. Here and in the following we chose gaussian kernels with bandwidth according to Scott's rule as implemented in `scipy.stats.gaussian_kde` (SciPy version 1.11.1). For the timer model, we then generated $(t, n)$ predictions by sampling $\lambda$ from $p(\lambda)$, $t$ from $p(t)$ and calculating $n = 2 \exp(\lambda t)$. For the counter model, we generated $(t, n)$ by sampling $\lambda$ from $p(\lambda)$, $n$ from $p(n)$ and calculating $t = \log(n/2)/\lambda$. All KDEs and predicted densities were discretized to integer $n$ and 30 min bins for $t$ to reproduce the binning of the experimental data.

### Bayesian model selection

We noticed that whenever the growth rate is tightly controlled, the two models generate indistinguishable $(t, n)$ scatter plots, as all data then collapse onto the same exponential growth curve. Our data nevertheless discriminate between models because growth rates exhibit variability. To quantify their relative support by data, we calculated Bayes factors (odds ratios)

$$K = \frac{\prod_i^N p(t_i, n_i | M_c)}{\prod_i^N p(t_i, n_i | M_t)},$$

where $M_{c,t}$ designate counter and timer models, respectively; $N$ is the number of data points, and $p(t, n|M)$ is the probability to observe a data point in time bin $t$ and with number $n$ within model $M$ as described above. Thus, $K > 1$ indicates a preference for the counter model, but only $K > 100$ is considered as "decisive" support. Because our models do not contain adjustable parameters, the usual integration over parameters is not needed here for the calculation of Bayes factors. To assess robustness to small sample variation, we bootstrap resampled $10^5$ size $N$ datasets, and then computed the fraction of corresponding $K$ values indicating "decisive" support for a model.

### Goodness-of-fit

A preferred model need not be a good fit for the data. To quantify fit quality, we calculated the Jensen-Shannon distance (JSD) between a model and the data, as follows. We first obtained a KDE $p_D(t, n)$ from the data and calculated its average with the model density $p_M(t, n) = p(t, n|M)$ as $\bar{p}(t, n) = 1/2[p_D(t, n) + p_M(t, n)]$. Then

$$\text{JSD} = (H[\bar{p}] - 1/2(H[p_D] + H[p_M]))^{1/2},$$

where the information entropy is given as a sum over time and number bins

$$H[p] = -\sum_{t,n} p(t, n)\log p(t, n).$$

The JSD is bounded between 0 (identical distributions) and 1 (nonoverlapping distributions). We then determined a $p$ value for a model $M$, by sampling $10^6$ synthetic size $N$ datasets from $M$, each time calculating JSD. This procedure accounts for the nonzero distances between data and model that are due do finite sample size only. We then evaluated the fraction $p$ of the synthetic JSDs exceeding the JSD between $M$ and the original dataset. Thus, $p \ll 1$ indicates a poor fit, whereas $p$ higher than a few percent indicates that the data are representative of a sample from $M$ and thus, a good fit.

## Supporting information

**S1 Fig. Schematics of plasmids used for parasite transfection. A** Two versions of Histone-2B-GFP expressing plasmids used for transfection of P. falciparum and P. knowlesi, respectively. **B** Two plasmids used for double transfection of P. falciparum for cell and nuclear volume analysis.
(TIF)

**S2 Fig. Parasite multiplication rate in P. falciparum and P. knowlesi is unaffected by episomal expression of different markers. A** Parasitemia of wild type and overexpression lines was assessed 48 h or 24 h apart for asynchronous P. falciparum or P. knowlesi cultures, respectively. 4 or 5 replicas were counted per condition. **B** Datapoints from A converted into multiplication rate per cycle are not different between P. falciparum or P. knowlesi lines. Black bars represent mean and SD. Statistical analysis: t-test with Welch's correction.
(TIF)

**S3 Fig. Quantification of schizont stage duration and number of daughter cells using microtubule marker in P. falciparum and P. knowlesi is reproducible. A** Airyscan-processed time-lapse images of P. falciparum strain 3D7 stained with SPY555-Tubulin. Timepoint of first spindle elongation was set to 0 h and the timepoint where segmentation started was considered as schizogony end. Shown are maximum intensity projections. Scale bar is 1 μm. **B** Same as A but for P. knowlesi strain A1-H.1. **C** Quantification of schizont stage duration in hours and **D** final merozoite number based on number of subpellicular microtubule structures. Black bars represent mean and SD. **E** Correlation of merozoite number against duration of schizont stage for P. falciparum. N = 12. Given are Pearson correlation coefficient r and p values. **F** Same as E but for P. knowlesi. N = 28.
(TIF)

**S4 Fig. Timer and counter model construction from data.** *A Growth rate λ vs. duration t (left) and vs. merozoite number n (right) for P. falciparum H2B-GFP expressing schizonts. The timer model density (blue) is constructed from the marginals for λ and t (inset, blue) assuming their independence. The counter model density (red) is constructed similarly from the marginals for λ and n (inset, red). B-D As (A) but for the other strains as indicated.*
(TIF)

**S5 Fig. Variability in merozoite number is reduced by the negative correlation between duration of schizont stage t and multiplication rate λ.** Merozoite number data as in [Fig 1E] but for all SPY555-Tubulin labelled and H2B-GFP expressing strains, alongside synthetic data generated by randomly reassigning t and λ pairs, followed by recalculating n according to the timer model. Shown are SD and quartiles, and shuffled data points as a swarm plot. Variation of shuffled data is higher as indicated by the interquartile ranges below.
(TIF)

**S6 Fig. Duration of schizont stage and merozoite number in P. falciparum expressing nuclear mCherry is consistent with H2B-GFP strain.** *A Quantification of schizont stage duration in hours and B merozoite number of P. falciparum 3D7 episomally expressing a nuclear mCherry signal (NLS-mCherry). Given are means and SEM. C Growth rate λ vs. duration t (left) and vs. merozoite number n (right) for P. falciparum NLS-mCherry and cytoplasmic GFP expressing schizonts. The timer model density (blue) is constructed from the marginals for λ and t (inset, blue) assuming their independence. The counter model density (red) is constructed similarly from the marginals for λ and n (inset, red). D Measured number vs. duration. Timer (blue, left) and counter (red, right) predict different densities in the t−n plane and marginal distribution*

*in n and t, respectively, based on the data. Jensen-Shannon distance (JSD) between data and model density (marker) for timer and counter (top and bottom, respectively). JSD histograms for synthetic data sampled from timer (blue) and counter (red) models show that the data are representative of a counter but not timer sample.*
(TIF)

**S7 Fig. Image segmentation of cytosolic and nucleoplasmic markers allows for volume calculations.** Exemplary single image slices from time points pre-schizogony ($t_{-2}$), schizogony start ($t_0$) and schizogony end ($t_{end}$) showing cytoplasmic marker pARL-GFP (green), nuclear marker NLS-mCherry (cyan), and thresholding masks generated for measuring respective areas and volumes. GFP mask was determined by automatic thresholding and mCherry by manual and visual adjustment. Shown is only z slice 11 of a stack of 20 slices to demonstrate segmentation, while analysis was carried out on the entire stack, which then contains all the nuclei. Contrast was adjusted for better visualization. Scale bar is 1 μm.
(TIF)

**S8 Fig. Correlations of t and λ against parasite cell volume at three timepoints of schizogony.** *Show are regression curves of cellular parameters* t *and* λ *measured in P. falciparum 3D7 episomally expressing a nuclear mCherry signal and cytoplasmic GFP plotted against parasite cell volume measured at replication start, schizont stage onset, and schizont stage end. N = 23 for* $t_{-2}$ *and N = 29 for all others from three independent replicates. Given are Pearson correlation coefficient r and p values. Values are bootstrapped to 95% confidence interval.*
(TIF)

**S9 Fig. Merozoite number does not correlate with N/C-ratio.** Correlation of merozoite number against N/C ratio all for **A** pre-schizogony ($t_{-2}$), **B** schizogony start ($t_0$) and **C** schizogony end ($t_{end}$). Given are Pearson correlation coefficient r and p values. N = 23 for $t_{-2}$ and N = 29 for all others.
(TIF)

**S10 Fig. P. knowlesi compacts its chromatin during schizogony. A** Chromatin volumes based on H2B-GFP marker for individual cells at three timepoints; pre-schizogony ($t_{-2}$), schizogony start ($t_0$) and schizogony end ($t_{end}$). Error bars represent mean and SEM. Statistics: t-test with Welch's correction. **B** Average nuclear volume (total chromatin volume divided by nuclear number) for individual cells. **C** Correlation of merozoite number against average nuclear volume at end of schizogony ($t_{end}$). Given are Pearson correlation coefficient r and p values. N = 24 for $t_{-2}$, N = 26 for $t_0$ and N = 25 for $t_{end}$.
(TIF)

**S11 Fig. Parasites can be cultured in medium diluted to 0.5x.** Growth curve with asynchronous parasite cultures with a slight majority of late stages cultivated in normal (1x, black) and diluted (0.5x, blue; 0.33x, purple; 0.25x, green) medium with 0.9% NaCl. Starting parasitemia was set around 0.2%. Medium was changed every 48h and parasitemia was assessed using Giemsa-stained thin blood smears. High parasitemias at the last time point might cause a reduction in growth dynamics for 1x and 0.5x medium, while 0.33x and 0.25x could never be maintained over prolonged time periods. Plotted are mean and SD of three technical replicates.
(TIF)

**S12 Fig. Medium dilution has no major effect on red blood cell indices.** Automated haematology analyses of erythrocytes after 24 and 48 h in different medium dilutions (1x, black; 0.5x, blue; 0.33x, purple; 0.25x, green). Plotted are the indices mean cell volume (MCV) in fL, mean

cell hemoglobin (MCH) in pg and mean cell hemoglobin concentration (MCHC) in g/dL. Shown are mean and SD of three technical replicates.
(TIF)

**S13 Fig. Predictions by timer and counter models compared for P. falciparum NLS in 0.5x diluted growth medium. A** As (S6C Fig) but for 0.5x medium. **B** As (S6D Fig, left subpanel) but for 0.5x medium. See also (Fig 6G and 6H).
(TIF)

**S1 Movie. H2B-GFP P. falciparum.** Airyscan-processed time-lapse imaging of P. falciparum 3D7 episomally expressing PfH2B tagged with GFP. Shown are images taken every hour during a total of 14 hours as maximum intensity projections of 21 slices in a 0.3 μm interval. Left, GFP; right, Brightfield and GFP (green). Image size is 15 x 15 μm.
(AVI)

**S2 Movie. H2B-GFP P. knowlesi.** Airyscan-processed time-lapse imaging of P. knowlesi A1-H.1 episomally expressing PkH2B tagged with GFP. Shown are images taken every 30 min during a total of 13.5 hours as maximum intensity projections of 20 slices in a 0.36 μm interval. Left, GFP; right, Brightfield and GFP (green). Image size is 12 x 12 μm.
(AVI)

**S3 Movie. SPY555-Tubulin P. falciparum.** Airyscan-processed time-lapse imaging of P. falciparum 3D7 stained with live Tubulin dye (SPY555-Tubulin, 1:2000). Shown are images taken every 30 min during a total of 14.5 hours as maximum intensity projections of 13 slices in a 0.5 μm interval. Left, Tubulin; right, Brightfield and Tubulin (magenta). Image size is 12 x 12 μm.
(AVI)

**S4 Movie. pARL-GFP NLS-mCherry P. falciparum.** Airyscan-processed time-lapse imaging of P. falciparum 3D7 episomally expressing cytosolic GFP (pARL-GFP) and nuclear mCherry (NLS-mCherry). Shown are images taken every 30 min during a total of 17 hours as maximum intensity projections of 20 slices in a 0.36 μm interval. Left, pARL-GFP (green), NLS-mCherry (magenta); right, Brightfield, pARL-GFP, and NLS-mCherry. Image size is 12 x 12 μm.
(AVI)

**S5 Movie. pARL-GFP NLS-mCherry P. falciparum in 0.5x diluted culture medium.** Airyscan-processed time-lapse imaging of P. falciparum 3D7 episomally expressing cytosolic GFP (pARL-GFP) and nuclear mCherry (NLS-mCherry). Acquisition started after about 24 hours incubation in diluted medium. Shown are images taken every 30 min during a total of 17 hours as maximum intensity projections of 20 slices in a 0.36 μm interval. Left, pARL-GFP (green), NLS-mCherry (magenta); right, Brightfield, pARL-GFP, and NLS-mCherry. Image size is 11 x 11 μm.
(AVI)

**S1 Text. ImageJ macro used for volume determination.**
(DOCX)

## Acknowledgments

We thank: The Infectious Diseases Imaging Platform for imaging support (idip-heidelberg. org). Markus Ganter for providing the NLS-mCherry strain and critical comments on the manuscript. Robert Moon for providing *P. knowlesi* acceptor strain with the Pkcon-GFP plasmid.

## Author Contributions

**Conceptualization:** Nils B. Becker, Julien Guizetti.

**Data curation:** Vanessa S. Stürmer, Sophie Stopper, Patrick Binder, Nils B. Becker.

**Formal analysis:** Vanessa S. Stürmer, Sophie Stopper, Patrick Binder, Nils B. Becker.

**Funding acquisition:** Nils B. Becker, Julien Guizetti.

**Investigation:** Vanessa S. Stürmer, Sophie Stopper, Anja Klemmer, Nicolas P. Lichti, Julien Guizetti.

**Methodology:** Vanessa S. Stürmer, Sophie Stopper, Patrick Binder, Anja Klemmer.

**Project administration:** Nils B. Becker, Julien Guizetti.

**Supervision:** Nils B. Becker, Julien Guizetti.

**Validation:** Vanessa S. Stürmer, Sophie Stopper.

**Visualization:** Vanessa S. Stürmer, Sophie Stopper, Patrick Binder, Nils B. Becker, Julien Guizetti.

**Writing – original draft:** Nils B. Becker, Julien Guizetti.

**Writing – review & editing:** Vanessa S. Stürmer, Patrick Binder, Nils B. Becker, Julien Guizetti.

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
