## [Decision Letter · Decision Letter 0]

7 Nov 2023

Dear Dr. Guizetti,

We are pleased to inform you that your manuscript 'Progeny counter mechanism in malaria parasites is linked to extracellular resources' has been provisionally accepted for publication in PLOS Pathogens.

Best regards,

Kami Kim

Section Editor

PLOS Pathogens

Kami Kim

Section Editor

PLOS Pathogens

Kasturi Haldar

Editor-in-Chief

PLOS Pathogens

orcid.org/0000-0001-5065-158X

Michael Malim

Editor-in-Chief

PLOS Pathogens

orcid.org/0000-0002-7699-2064

Reviewer Comments (if any, and for reference):

Reviewer's Responses to Questions

**Part I - Summary**

Reviewer #1: I will keep this summary brief, as each of the three Review Commons reviewers have already provided their own summaries. The authors provide an exemplary response to the comments of their three reviewers. Essentially all points raised by the reviewers are thoroughly covered.

This study represents both technological advances in studying the replication of malaria parasites, and also what in my opinion is a conceptual advancement in regards to thinking about how parasites replicate. Despite its utility in other fields, very few previous studies have applied mathematical modelling to biological questions of parasite replication.

I raise a single concern on behalf of one of the initial reviewers about validation of genetic constructs used in this study. This should be very easily addressed by the authors, at which point the manuscript will in my opinion be fit and appropriate for publication.

Reviewer #2: Stürmer and colleagues used super-resolution time-lapse microscopy to probe the mechanism regulating the number of merozoites produced by a single cell in Plasmodium falciparum and P. knowlesi. By fitting these in to a statistical model the authors conclude the follwoings:

a. P. knowlesi has a similar duration of schizont stage to P. falciparum, although it has a shorter intraerythrocytic developmental cycle (IDC).

b. Nuclear multiplication dynamics in P. knowlesi suggest a counter mechanism of division, which is further supported by a significant correlation between merozoite number and schizont size at the onset of division.

c. Nutritional deprivation in P. falciparum causes an increase in nuclear volume and a decrease in merozoite number.

The main innovation of this work was the use of high-resolution live time-lapse microscopy to monitor nuclei division using new cell division markers and combining these data with a statistical model. All these data bolden their claims.

The readership of this paper is very limited though- it will be of interest to the apicomplexan biologists and, perhaps most fittingly, to theorists of cell biology.

Reviewer #3: I don't agree with all of the conclusions drawn in this paper, but I nonetheless regard them as reasonable and thoughtfully argued, and the revision has adequately addressed my major concerns.

**Part II – Major Issues: Key Experiments Required for Acceptance**

Reviewer #1: I mirror a concern raised by the initial Reivewer # 3 about genetic validation of the genetic constructs used in this study. While I agree with the authors that resistance to selectable markers and distinct fluorescent markers are sufficient to validate episomal transfectants, the authors do not show a lack of fluorescence in the parental lines. In Supplementary Figure 1, the authors should either include genetic validation of these parasite lines as suggested by Reviewer # 3, or include comparison images of the fluorescence and lack of fluorescence in transfectants vs parental lines.

Reviewer #2: The authors responded to all my previous comments, as well as the comments to other reviewers. I do not have any further suggestions to make on the experimental side. However, I do not have sufficient expertise to comment on their statistical model presented in fig 2.

Reviewer #3: I consider the authors' responses to requests to be well argued, and I have no further suggested experiments that I consider necessary for acceptance.

**Part III – Minor Issues: Editorial and Data Presentation Modifications**

Reviewer #1: (No Response)

Reviewer #2: The authors responded to all my previous comments, as well as the comments to other reviewers. I do not have any further suggestions to make.

Reviewer #3: nil

PLOS authors have the option to publish the peer review history of their article (what does this mean?). If published, this will include your full peer review and any attached files.

Reviewer #1: **Yes: **Benjamin Liffner

Reviewer #2: **Yes: **Rubayet Elahi

Reviewer #3: No

---

## [Editor Report · Acceptance letter]

30 Nov 2023

Dear Dr. Guizetti,

We are delighted to inform you that your manuscript, "Progeny counter mechanism in malaria parasites is linked to extracellular resources," has been formally accepted for publication in PLOS Pathogens.

Best regards,

Michael Malim

Editor-in-Chief

PLOS Pathogens

orcid.org/0000-0002-7699-2064